# NoiseSDF2NoiseSDF: Learning Clean Neural Fields from Noisy Supervision

## Abstract

Reconstructing accurate implicit surface representations from point clouds remains a challenging task, particularly when data is captured using low-quality scanning devices. These point clouds often contain substantial noise, leading to inaccurate surface reconstructions. Inspired by the Noise2Noise paradigm for 2D images, we introduce NoiseSDF2NoiseSDF, a novel method designed to extend this concept to 3D neural fields. Our approach enables learning clean neural SDFs directly from noisy point clouds through noisy supervision by minimizing the MSE loss between noisy SDF representations, allowing the network to implicitly denoise and refine surface estimations. We evaluate the effectiveness of NoiseSDF2NoiseSDF on benchmarks, including the ShapeNet, ABC, Famous, and Real datasets. Experimental results demonstrate that our framework significantly improves surface reconstruction quality from noisy inputs.

## 1 Introduction

Learning from imperfect targets [48, 18, 24, 5, 50] is a fundamental challenge in machine learning, particularly when obtaining clean training labels is impractical or unfeasible. In image processing, the pioneering work of Noise2Noise [24] demonstrated that image restoration could effectively be achieved by observing multiple corrupted instances of the same scene. Specifically, this method leverages the principle that pixel values at identical coordinates in different noisy images ideally represent the same underlying true signal. Consequently, the model learns to restore clean images by simply minimizing a straightforward MSE loss between noisy observations. See Figure 1 (a).

Extending Noise2Noise principles to 3D point clouds [19, 30], however, poses significant challenges due to their inherently unstructured nature. Unlike images organized on regular grids, point clouds exhibit deviations across all spatial coordinates without the benefit of a stable reference framework. This fundamental difference renders a direct extension of unsupervised image denoisers impractical. Standard loss functions such as Mean Squared Error (MSE) prove ineffective, necessitating specialized loss functions like Earth Mover's Distance (EMD) to capture geometric correspondences and spatial distributions inherent in point cloud data, see Figure 1 (b).

Recent advances in surface reconstruction have introduced neural fields, such as neural Signed Distance Function (neural SDF) [34, 31], which are capable of predicting continuous SDF values for any given 3D coordinate. Our key observation is that neural SDF, which encodes the SDF mapping 3D coordinates to scalar distance values for 3D shape, exhibits a conceptual parallel to the mapping between pixel coordinates and pixel intensities in 2D images, as shown in Figure 1 (c). Building on this analogy, we hypothesize that neural SDFs can be denoised by directly using noisy SDF observations with the same MSE loss strategy inspired by the Noise2Noise principle in image restoration.

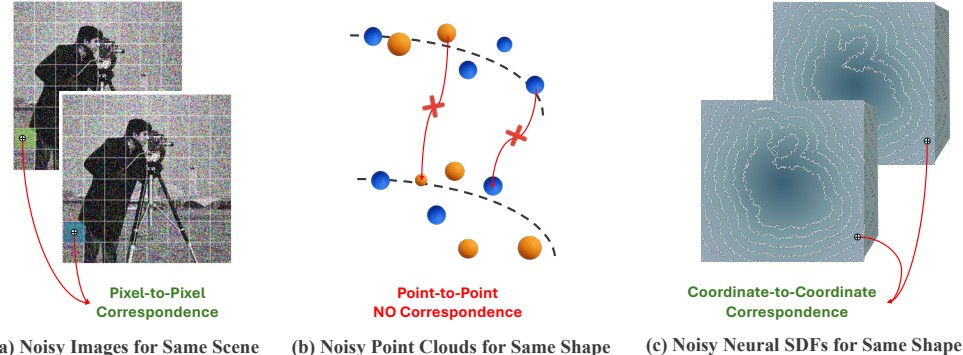

(a) Noisy Images for Same Scene    (b) Noisy Point Clouds for Same Shape    (c) Noisy Neural SDFs for Same Shape

**Figure 1:** Comparison of coordinate correspondences: (a) Pixel coordinates represent correspondences between two noisy images of the same scene. In contrast, (b) point coordinates do not exhibit correspondences between two noisy point clouds of the same shape. (c) SDF coordinates establish correspondences between two noisy neural fields representing the same shape.

In this work, we explore the use of noisy-target supervision in neural SDFs for surface reconstruction from noisy 3D point clouds. We propose NoiseSDF2NoiseSDF, an adaptation of the Noise2Noise framework applied to neural fields. The workflow of our NoiseSDF2NoiseSDF is illustrated in Figure 2. The network first takes independently corrupted point clouds as input to predict the underlying clean SDF values. Instead of using clean SDFs as ground truth, we employ another noisy neural SDF, which is generated by off-the-shelf point-to-SDF methods, as the supervision target. We then minimize the discrepancy between the predicted SDF output and the noisy SDF target using MSE loss. The network learns to suppress noise and improve consistency across SDF values, leading to clean neural representations.

To evaluate the effectiveness of NoiseSDF2NoiseSDF, we conduct comprehensive experiments across benchmark datasets, including ShapeNet [8], ABC [22], Famous [14], and Real [14]. Our experimental results demonstrate that neural SDFs can indeed be denoised effectively by employing mean squared error loss directly between their noisy representations. This finding confirms our central hypothesis: neural SDFs can learn to produce cleaner outputs simply by observing and minimizing discrepancies among noisy neural fields, effectively extending the Noise2Noise paradigm into the domain of 3D neural surface reconstruction. Our approach eliminates the need for clean training data, making it practical and scalable for real-world scenarios where acquiring perfect data is difficult or infeasible.

## 2 Related Work

**Noise2Noise.** The Noise2Noise (N2N) framework [24] has significantly influenced recent image denoising. By leveraging pairs of noisy observations of the same scene, the N2N framework learns to predict one noisy realization from another via pixel-wise correspondence. Subsequent methods like Noise2Void [23], Noise2Self [3] employ blind-spot masking techniques, training models directly on individual noisy images without pairs. Noise2Same [43] derives self-supervised loss bounds to eliminate the blind-spot restriction altogether. Self2Self [38] and Neighbor2Neighbor [20] exploit internal image redundancy, employing dropout or pixel resampling to train directly on single noisy observations without explicit noise modeling. Noisier2Noise [33] extends N2N to explicitly introduce additional synthetic noise, learning to map noisier images back to their original noisy versions.

Extending the Noise2Noise framework to 3D [19, 42, 30, 40] is challenging due to the unordered nature of point clouds. Methods like TotalDenoising [19] and Noise2Noise Mapping [30] address this by leveraging local geometric correspondences instead of exact point matches, replacing MSE with Earth Mover's Distance (EMD) loss to better align noisy point clouds with the underlying surface. In our work, we exploit the structural similarities between neural fields and images, proposing a Noise2Noise denoising framework for 3D SDFs with MSE loss.

**Implicit Surface Reconstruction.** Learning implicit surfaces from point clouds has seen significant advances. Overfitting-based methods optimize a neural implicit function for a single point cloud,

often with ground-truth SDFs, normals or geometric constraints and physical priors. For example, SAL [1], SALD [2], and Sign-SAL [49] use point proximity and self-similarity cues. Gradient regularization techniques like IGR [17], DiGS [4], and Neural-Pull [29] improve stability and detail. Extensions such as SAP [36], LPI [9], and Implicit Filtering-Net [25] enhance reconstruction under sparse sampling and complex geometry. While accurate, these methods are typically sensitive to noise. Robust variants (e.g., SAP [36], PGR [26], Neural-IMLS [41], Noise2Noise Mapping [30] and LocalN2NM [10]) address this via smoothing, denoising priors, or self-supervision.

In contrast to overfitting approaches, data-driven methods learn shape priors from large datasets. Global-latent methods, such as OCCNet [31], IM-NET [11], and DeepSDF [34], encode entire shapes into fixed-length latent codes, capturing overall semantics but often over-smoothing details. Local prior methods improve expressiveness by operating at finer scales. Grid-based approaches divide space into cells and learn small implicit functions per cell (ConvOccNet [35], SSRNet [32], Local Implicit Grid [16], Deep Local Shapes [7]). Patch-based methods segment point clouds into local regions and learn shared atomic representations (PatchNets [39], POCO [6], neighborhood-based [21]). Hybrid methods combine global context with local detail. For instance, IF-Nets [12] and SG-NN [13] fuse PointNet features with voxel hierarchies or contrastive scene priors. Recent transformer-based models (ShapeFormer [44], 3DILG [46], 3DS2V [47], LaGeM [45]) leverage self-attention for long-range structure modeling. Data-driven models trained with ground-truth SDFs are generally more robust to noise. Hybrid approaches like P2S [37] and PPSurf [15] use dual-branch networks to predict SDFs with explicit noise-level supervision. However, their performance degrades under extreme sparsity and noise of input point clouds. In contrast, our NoiseSDF2NoiseSDF framework embraces noise as a training signal, enabling reliable surface recovery from severely degraded inputs.

# 3   Preliminaries

In *Noise2Noise* [24], the key idea is that given multiple noisy observations of the same underlying clean image, the pixel intensities at the same spatial coordinates are expected to share the same statistical properties. Formally, consider an image domain $\mathbf{X} \subset \mathbb{R}^2$, and let $y_1, y_2, \ldots, y_n$ be noisy observations of the same underlying clean image taken at different instances. For any pixel coordinate $x \in \mathbf{X}$, the pixel intensities $y_1(x), y_2(x), \ldots, y_n(x)$ are samples drawn from a distribution centered around the true pixel value at that location, perturbed by independent, zero-mean noise. The core insight of Noise2Noise is that even in the presence of such noise, the expectation of the noisy pixel values converges to the true signal:

$$\mathbb{E}[y_i(x)] = y(x), \quad \forall i \in \{1, 2, \ldots, n\}, \tag{1}$$

where $y_i(x)$ is the observed pixel value at coordinate $x$ in the $i$-th noisy image, and $y(x)$ is the true underlying pixel value at that coordinate. This property enables training a neural network purely on noisy data, using other noisy images as supervision.

Let $f_\theta$ denote a neural network parameterized by $\theta$, and let $x \in \mathbf{X}$ represent a spatial query coordinate. The network is designed to predict pixel intensities given a noisy image and the query coordinate. The prediction is written as:

$$\hat{y}(x \mid y_i) = f_\theta(y_i, x), \tag{2}$$

where $y_i$ is the noisy input image, $x$ is the queried pixel location, and $\hat{y}(x \mid y_i)$ is the predicted pixel intensity at $x$.

The model is trained to minimize the expected squared error between the predicted pixel value and the corresponding pixel value in another independent noisy observation. The loss function is:

$$\mathcal{L}(\theta) = \mathbb{E}_{y_1, y_2 \sim p(y|y), x \sim \mathcal{U}(\mathbb{R}^2)} \left[ \|\hat{y}(x \mid y_1) - y_2(x)\|^2 \right], \tag{3}$$

where $y_1, y_2$ are independent noisy observations of the same clean image, and $x \in \mathbf{X}$ is sampled uniformly from the image domain.

## 4 Method

Our proposed method investigates whether clean neural fields can be effectively learned by observing their noisy counterparts. Drawing inspiration from Noise2Noise, where noisy images directly serve as inputs and targets, we adapt this principle to learning neural fields from noisy point cloud data. In contrast to the direct usage of noisy images as input in traditional Noise2Noise setups, we employ a neural network conditioned on a noisy point cloud to predict neural SDFs at specific query coordinates. Rather than utilizing clean SDFs as supervision, our approach leverages noisy neural fields at identical coordinates derived from another independently noisy version of the same underlying shape. This ensures one-to-one correspondence between the predicted and target neural fields, allowing effective noise suppression through direct MSE loss minimization.

### 4.1 NoiseSDF2NoiseSDF

Applying *Noise2Noise* [24] to Signed Distance Functions (SDFs) introduces new opportunities for denoising in 3D spaces. Unlike unstructured point clouds, SDFs represent 3D geometry in a structured and continuous manner, mapping each spatial coordinate $q \in \mathbb{R}^3$ to its signed distance from the surface of an underlying object. This continuity ensures that, for the same query coordinate across multiple noisy observations derived from the same shape, the SDF values should remain statistically consistent. Let $p_1, p_2, \ldots, p_n$ be noisy point cloud observations of the same underlying 3D shape, and let $s_1, s_2, \ldots, s_n$ be their corresponding noisy SDFs. Given a noisy point cloud $p_i$ and a query coordinate $q$, a neural network $f_\theta$, parameterized by $\theta$, is trained to predict the SDF value $\hat{s}(q \mid p_i)$ at the queried location:

$$\hat{s}(q \mid p_i) = f_\theta(p_i, q). \tag{4}$$

The structured nature of SDFs enables the network to learn smooth and continuous surface representations, even from sparse or noisy inputs. This makes SDFs advantageous over unordered point clouds for tasks like 3D denoising and reconstruction.

**Training Objective.** The model is trained by minimizing the expected squared error between the predicted SDF value from one noisy observation and the SDF value at the same query location in another noisy observation of the same shape. The loss function is defined as:

$$\mathcal{L}(\theta) = \mathbb{E}_{p_1, p_2 \sim p(p|s), q \sim \mathcal{U}(\mathbb{R}^3)} \left[ \|\hat{s}(q \mid p_1) - s_2(q \mid p_2)\|^2 \right], \tag{5}$$

where $p_1, p_2$ are independent noisy point cloud observations sampled from the same underlying shape $s$, and $s_2(q \mid p_2)$ is the noisy SDF value at coordinate $q$ associated with noisy point cloud $p_2$.

This formulation takes advantage of the continuous nature of SDFs, which, unlike point clouds, allows for consistent supervision across noisy samples even if the raw point distributions are unstructured. By learning to map noisy coordinates to structured SDF representations, the neural network effectively filters noise, yielding a refined and more accurate 3D representation of the surface.

### 4.2 Implementation

Our framework is illustrated in Figure 2. The process begins with sampling sparse, noisy point clouds from a watertight surface. During training, a pair of noisy point clouds is randomly selected: one is processed through the neural SDF network to predict approximate clean SDF values for the underlying 3D shape. Simultaneously, a point-to-SDF method is applied to generate a noisy SDF target, which serves as noisy supervision during the denoising phase. For each query point, the corresponding SDF values from these two representations are extracted and compared using the Mean Squared Error (MSE) loss function. This loss is then utilized to update the weights of the neural SDF network during denoising.

**Point Sampling.** We first normalize the watertight meshes into a unit cube, then sample points from the surfaces to obtain the original point cloud $p$. Following the Noise2Noise Mapping protocol [24, 30], we apply zero-mean Gaussian noise to generate noisy point cloud pairs. The query point set consists of $50\%$ near-surface points and $50\%$ uniformly sampled points from the unit cube. To reduce

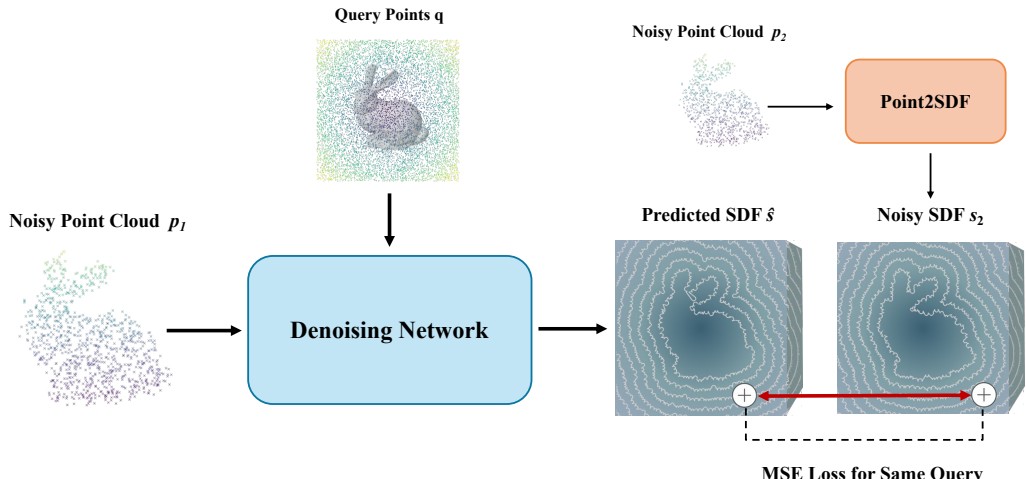

**Figure 2:** The training pipeline of the NoiseSDF2NoiseSDF framework. Given two independent noisy point clouds $p_1$ and $p_2$ of the same underlying shape, $p_1$ is fed into the denoising network to predict a smoothed SDF $\hat{s}$, while $p_2$ is passed through a Point2SDF network to generate a noisy SDF $s_2$. Both SDFs are evaluated at a shared set of query points $q$, and their mean squared error is used to update the denoising network weights.

dependency on the original clean surface, we directly use the two input noisy point clouds as the near-surface query points. Additionally, we uniformly sample $N$ points within the cube as spatial query points.

**Denoising Network.** Our SDF prediction network is built on 3DS2V [47]. Initially, a noisy point cloud $p_1$ is sampled and transformed into positional embeddings, which are then encoded into a set of latent codes through a cross-attention module. Subsequently, self-attention is applied to aggregate and exchange information across the latent set, enhancing feature integration. A cross-attention module then computes interpolation weights for the query point $q$. These interpolated feature vectors are processed through a fully connected layer to predict SDF values. The network is initialized using pretrained weights from 3DS2V. Without the denoising learning, the predicted SDF values are typically noisy due to the inherent noise in the input point clouds. During our denoising phase, the encoder remains fixed while only the decoder is optimized, improving both efficiency and convergence speed. Importantly, our contribution is not limited to the specific architecture of the SDF prediction network; other point-to-SDF methods could also be utilized.

**Noisy Target.** Given another paired noisy point cloud $p_2$, a Point2SDF method is required to predict noisy SDF values $s_2$ from it. In our implementation, we also employ a pretrained 3DS2V [47] model. This network accepts as input a noisy point cloud $p_2$ and a query point $q$, producing the corresponding noisy SDF scalar value at $q$. To ensure that all SDF targets are consistently noisy, we freeze its parameters during this process. Notably, this Point2SDF method is also not restricted to a specific model; any data-driven or overfitting Point2SDF methods could be seamlessly integrated. The Mean Squared Error (MSE) loss function is employed to minimize discrepancies between corresponding SDF values. This loss guides the optimization of the neural SDF network's weights during the denoising training.

## 5 Experiment

We designed experiments to evaluate the performance of the NoiseSDF2NoiseSDF framework for surface reconstruction from raw noisy point clouds, assessing performance across different noise levels using a data-driven paradigm on large 3D shape datasets. We conducted ablation studies to validate key components and design choices.

## 5.1 Training Details

For optimization, we used the AdamW optimizer [28] with a fixed learning rate of $1 \times 10^{-4}$. For resource usage, we trained on three Nvidia A100 GPUs with a batch size of 32 per GPU, taking approximately 15 hours for the ShapeNet dataset and 2.5 hours for the ABC dataset.

We sampled 2048 points from watertight meshes as the initial point cloud. Following the Noise2Noise Mapping [30], we applied Gaussian noise with standard deviations of $1\%$, $2\%$, online to generate noisy and sparse point cloud pairs. Additionally, we sampled 8192 query points online. The noise magnitude is defined with respect to both the point-cloud bounding-box size and the point density. For a fixed numeric noise level, a smaller bounding box amplifies the relative impact of the perturbation. All point clouds are normalized to the cubes $[-0.5, 0.5]^3$ or $[-1, 1]^3$. Furthermore, sparser point sets are more susceptible to noise. With only 2048 points, noise levels of $0.01$ and $0.02$ constitute *severe* perturbations irrespective of the bounding-box scale. The clean underlying surface is recovered from the denoised SDF using the Marching Cubes [27].

## 5.2 Datasets and Metrics

We trained our NoiseSDF2NoiseSDF network on the ShapeNet dataset following [47], using the same data split and preprocessing procedures. To evaluate denoising effectiveness and surface reconstruction quality, we used several metrics, including Intersection-over-Union (IoU), Chamfer Distance, F1 Score, and Normal Consistency.

IoU was computed based on occupancy predictions over densely sampled volumetric points. Following methods [25, 29], we sampled $1 \times 10^5$ points from the reconstructed and ground-truth surfaces to compute the Chamfer Distance and F1 Score.

To further assess the generalization capability of NoiseSDF2NoiseSDF, we further trained our model on the ABC train set [22]—which was not used during the pretraining of 3DS2V—and then evaluated it on the ABC test set, as well as the Famous [14] and Real [14] datasets. We utilized the preprocessed datasets and data splits provided by [14, 15]. We reported evaluation metrics including Normal Consistency, Mesh Normal Consistency, Chamfer Distance, and F1 Score. All metrics reported above are evaluated on the reconstructed meshes. We excluded IoU from this benchmark because, under severe noise, many reconstructed meshes become non-watertight or heavily degenerated, making it infeasible to assign reliable inside/outside labels and rendering the IoU metric unreliable.

## 5.3 Results on ShapeNet

We compared our method against 3DS2V [47] and 3DILG [46] on the seven largest ShapeNet subsets, using the same training and testing data split. We used the official pretrained models released by the respective authors. Evaluation results are reported for each subset at noise levels of 0.01 (Table 1) and 0.02 (Table 2); see visualization results in Figure 3. The results demonstrate that our method maintains robustness under both mild $\sigma = 0.01$ and severe $\sigma = 0.02$ corruption levels. Under lower corruption ($\sigma = 0.01$), our method outperforms all competing methods across all evaluation metrics. For instance, IoU rises by 4% for chair and by about 9% for rifle. Chamfer Distance drops from 0.010 to 0.008 for airplane and from 0.011 to 0.009 for lamp. Normal Consistency and F-Score likewise see significant improvements. At the higher corruption level ($\sigma = 0.02$), while the performance of all methods degrades, our approach remains the most robust and stable with the best mean metrics surpassing all baseline models. In challenging categories such as table, rifle, and lamp, our model still leads: rifle achieves an IoU of 0.781 and a Chamfer Distance of 0.014.

## 5.4 Results on ABC, Famous, and Real

We compared results on the ABC, Famous, and Real test datasets offered by P2S [14]. We compared three data-driven methods P2S [14], PPSurf [15], and 3DS2V [47] and two overfitting methods SAP-O [36], PGR [26]. These methods are widely recognized for their strong resilience to noise in point cloud data. For the data-driven methods, we used the pretrained models provided by the original authors. For the overfitting baselines we adopted the training configurations recommended or set as default in their respective works. Quantitative results are reported in Table 3 and Table 4, and qualitative mesh reconstructions are visualized in Figure 4.

**Table 1:** Performance comparison of 3DS2V [47], 3DILG [46], and Ours on ShapeNet test datasets derived from the 3DS2V with an additional Gaussian noise $\sigma = 0.01$. Higher is better for IoU, NC, and F-Score; lower is better for Chamfer. Best results are highlighted in bold.

| Category | IoU ↑ | | | NC ↑ | | | Chamfer ↓ | | | F-Score ↑ | | |
|---|---|---|---|---|---|---|---|---|---|---|---|---|
| | 3DS2V | 3DILG | Ours | 3DS2V | 3DILG | Ours | 3DS2V | 3DILG | Ours | 3DS2V | 3DILG | Ours |
| table | 0.879 | 0.856 | **0.922** | 0.930 | 0.932 | **0.976** | 0.013 | 0.014 | **0.012** | 0.991 | 0.982 | **0.992** |
| car | 0.946 | 0.931 | **0.959** | 0.890 | 0.861 | **0.908** | 0.022 | 0.025 | **0.020** | 0.925 | 0.899 | **0.925** |
| chair | 0.887 | 0.881 | **0.921** | 0.937 | 0.930 | **0.966** | 0.014 | 0.015 | **0.013** | 0.986 | 0.977 | **0.986** |
| airplane | 0.884 | 0.871 | **0.931** | 0.939 | 0.924 | **0.972** | 0.010 | 0.010 | **0.008** | 0.997 | 0.990 | **0.997** |
| sofa | 0.946 | 0.942 | **0.964** | 0.943 | 0.934 | **0.974** | 0.014 | 0.015 | **0.012** | 0.986 | 0.980 | **0.987** |
| rifle | 0.821 | 0.839 | **0.910** | 0.869 | 0.875 | **0.960** | 0.009 | 0.010 | **0.007** | 0.997 | 0.991 | **0.998** |
| lamp | 0.826 | 0.825 | **0.894** | 0.904 | 0.883 | **0.952** | 0.011 | 0.016 | **0.009** | 0.989 | 0.956 | **0.989** |
| mean | 0.884 | 0.878 | **0.929** | 0.916 | 0.905 | **0.958** | 0.0132 | 0.015 | **0.0113** | 0.981 | 0.968 | **0.986** |

**Table 2:** Performance comparison of 3DS2V [47], 3DILG [46], and Ours under an additional Gaussian noise $\sigma = 0.02$. Best results are highlighted in bold.

| Category | IoU ↑ | | | NC ↑ | | | Chamfer ↓ | | | F-Score ↑ | | |
|---|---|---|---|---|---|---|---|---|---|---|---|---|
| | 3DS2V | 3DILG | Ours | 3DS2V | 3DILG | Ours | 3DS2V | 3DILG | Ours | 3DS2V | 3DILG | Ours |
| table | 0.528 | 0.430 | **0.591** | 0.765 | 0.756 | **0.912** | 0.029 | 0.036 | **0.028** | 0.792 | 0.723 | **0.859** |
| car | 0.434 | **0.541** | 0.491 | 0.715 | 0.699 | **0.787** | **0.040** | 0.052 | 0.044 | 0.669 | 0.600 | **0.688** |
| chair | 0.463 | 0.392 | **0.530** | 0.729 | 0.712 | **0.868** | **0.034** | 0.037 | 0.035 | 0.694 | 0.701 | **0.721** |
| airplane | 0.465 | **0.564** | 0.536 | 0.719 | 0.710 | **0.856** | 0.025 | 0.031 | **0.022** | 0.830 | 0.764 | **0.899** |
| sofa | 0.355 | 0.312 | **0.425** | 0.769 | 0.737 | **0.866** | **0.036** | 0.039 | 0.038 | 0.667 | 0.664 | **0.677** |
| rifle | 0.625 | 0.564 | **0.781** | 0.691 | 0.691 | **0.891** | 0.021 | 0.029 | **0.014** | 0.887 | 0.776 | **0.968** |
| lamp | 0.572 | 0.461 | **0.649** | 0.744 | 0.712 | **0.896** | 0.026 | 0.040 | **0.025** | 0.825 | 0.699 | **0.880** |
| mean | 0.492 | 0.466 | **0.572** | 0.733 | 0.717 | **0.868** | 0.030 | 0.038 | **0.029** | 0.766 | 0.704 | **0.813** |

*For data-driven comparison*, across all noise levels our method achieves the highest NC and Mesh NC, indicating the most coherent geometry and the smoothest surfaces; this is also evident in the visual reconstructions (Figure 4). At the 0.01 noise level, the ABC and Famous datasets reach NC scores of 0.865 and 0.831, respectively—an improvement of roughly 1.5–2.5 percentage points over the second-best approach. When the noise level increases to 0.02, all baselines degrade, yet our NC remains the best among them. *For methods that fit a surface to each test cloud individually*, Table 4 shows that SAP-O and PGR can sometimes obtain lower Chamfer distance and higher F1, but our approach still leads on NC and Mesh NC. Under 0.01 noise level, our average NC/ Mesh NC / F-Score reach 0.847/0.027/0.945, all of which are the top scores. However, when the noise level rises to 0.02, our Chamfer and F1 decrease substantially. This drop is partly due to the backbone 3DS2V model's sensitivity to heavy noise: its F1 plunges from 0.962 to 0.813 as the noise level increases from 0.01 to 0.02. As illustrated in Figure 4, the mesh geometry becomes severely corrupted and large holes appear. Although NoiseSDF2NoiseSDF averages the neural SDF and closes some gaps, it cannot perfectly recover the underlying surface. Moreover, the smoothing effect of our model at the 0.02 noise level removes certain fine details; while this yields visually smoother meshes, Chamfer distance and F1—metrics that emphasize global geometric fidelity rather than surface roughness—are consequently worse.

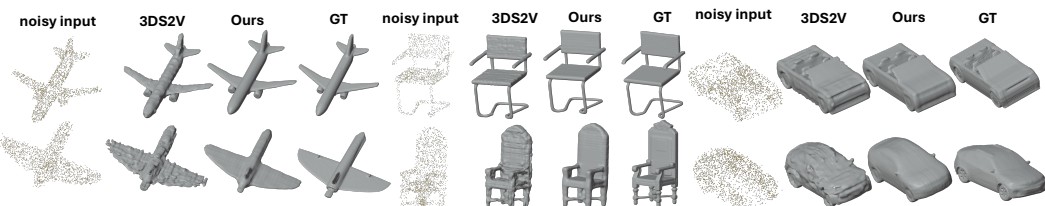

**Figure 3:** Comparison on the ShapeNet dataset. The first row corresponds to Gaussian noise with standard deviation $\sigma = 0.01$, and the second row to $\sigma = 0.02$. Compared to the baseline 3DS2V[47] method, our approach produces smoother reconstructions that more closely align with the underlying surfaces.

**Table 3:** Comparison of P2S [14], PPSurf [15], 3DS2V [47], and Ours on six noisy test datasets. Higher is better for NC, F-Score; lower is better for Mesh NC and Chamfer distance.

| Dataset | NC ↑ | | | | Mesh NC ↓ | | | | Chamfer | | | | F-Score | | | |
|---|---|---|---|---|---|---|---|---|---|---|---|---|---|---|---|---|
| | P2S | PPSurf | 3DS2V | Ours | P2S | PPSurf | 3DS2V | Ours | P2S | PPSurf | 3DS2V | Ours | P2S | PPSurf | 3DS2V | Ours |
| ABC ($\sigma = 0.01$) | 0.790 | 0.770 | 0.859 | **0.865** | 0.330 | 0.059 | 0.036 | **0.024** | 0.017 | 0.017 | **0.014** | 0.015 | 0.919 | 0.935 | **0.959** | 0.938 |
| ABC ($\sigma = 0.02$) | 0.753 | 0.728 | 0.735 | **0.812** | 0.381 | 0.061 | 0.060 | **0.018** | 0.027 | **0.022** | 0.028 | 0.032 | 0.852 | **0.870** | 0.780 | 0.724 |
| Famous ($\sigma = 0.01$) | 0.771 | 0.761 | 0.819 | **0.831** | 0.268 | 0.053 | 0.040 | **0.025** | 0.017 | **0.015** | 0.014 | 0.016 | 0.928 | **0.959** | 0.962 | 0.941 |
| Famous ($\sigma = 0.02$) | 0.727 | 0.728 | 0.705 | **0.767** | 0.328 | 0.054 | 0.064 | **0.024** | 0.022 | **0.020** | 0.028 | 0.032 | 0.868 | **0.899** | 0.788 | 0.726 |
| Real ($\sigma = 0.01$) | 0.789 | 0.776 | 0.818 | **0.845** | 0.177 | 0.057 | 0.056 | **0.031** | 0.016 | 0.016 | **0.014** | 0.015 | 0.946 | 0.954 | **0.964** | 0.956 |
| Real ($\sigma = 0.02$) | 0.734 | 0.745 | 0.735 | **0.793** | 0.269 | 0.053 | 0.071 | **0.020** | **0.021** | 0.022 | 0.021 | 0.026 | **0.877** | 0.876 | 0.873 | 0.809 |
| mean ($\sigma = 0.01$) | 0.783 | 0.769 | 0.832 | **0.847** | 0.258 | 0.056 | 0.044 | **0.027** | 0.017 | 0.016 | **0.014** | 0.015 | 0.931 | 0.949 | **0.962** | 0.945 |
| mean (all) | 0.761 | 0.751 | 0.778 | **0.819** | 0.292 | 0.056 | 0.055 | **0.024** | 0.020 | **0.019** | 0.020 | 0.023 | 0.898 | **0.916** | 0.888 | 0.849 |

**Table 4:** Comparison of SAP-O [36], PGR [26], and Ours on six noisy test datasets. The released PGR implementation uses an adaptive Marching Cubes resolution that, for 2,048-point-point clouds, occasionally drops to 64 rather than 128, which can yield artificially smoother meshes.

| Dataset | NC ↑ | | | Mesh NC ↓ | | | Chamfer | | | F-Score | | |
|---|---|---|---|---|---|---|---|---|---|---|---|---|
| | SAP-O | PGR | Ours | SAP-O | PGR | Ours | SAP-O | PGR | Ours | SAP-O | PGR | Ours |
| ABC ($\sigma = 0.01$) | 0.710 | 0.835 | **0.865** | 0.079 | 0.037 | **0.024** | 0.021 | 0.020 | **0.014** | 0.906 | 0.896 | **0.938** |
| ABC ($\sigma = 0.02$) | 0.622 | 0.778 | **0.812** | 0.095 | 0.065 | **0.018** | 0.026 | **0.026** | 0.032 | **0.824** | 0.815 | 0.724 |
| Famous ($\sigma = 0.01$) | 0.745 | 0.813 | **0.831** | 0.053 | 0.035 | **0.025** | 0.022 | 0.017 | **0.016** | 0.876 | 0.931 | **0.941** |
| Famous ($\sigma = 0.02$) | 0.614 | 0.755 | **0.767** | 0.104 | 0.064 | **0.024** | **0.023** | 0.024 | 0.032 | 0.849 | 0.834 | **0.726** |
| Real ($\sigma = 0.01$) | 0.683 | 0.827 | **0.845** | 0.097 | 0.032 | **0.031** | 0.023 | 0.015 | **0.015** | 0.902 | 0.956 | **0.956** |
| Real ($\sigma = 0.02$) | 0.595 | 0.756 | **0.793** | 0.122 | 0.062 | **0.020** | 0.025 | 0.026 | 0.026 | **0.841** | 0.824 | 0.809 |
| mean ($\sigma = 0.01$) | 0.713 | 0.825 | **0.847** | 0.076 | 0.035 | **0.027** | 0.022 | 0.017 | **0.015** | 0.895 | 0.928 | **0.945** |
| mean (all) | 0.661 | 0.794 | **0.819** | 0.092 | 0.049 | **0.024** | 0.023 | **0.021** | 0.023 | 0.866 | **0.876** | 0.849 |

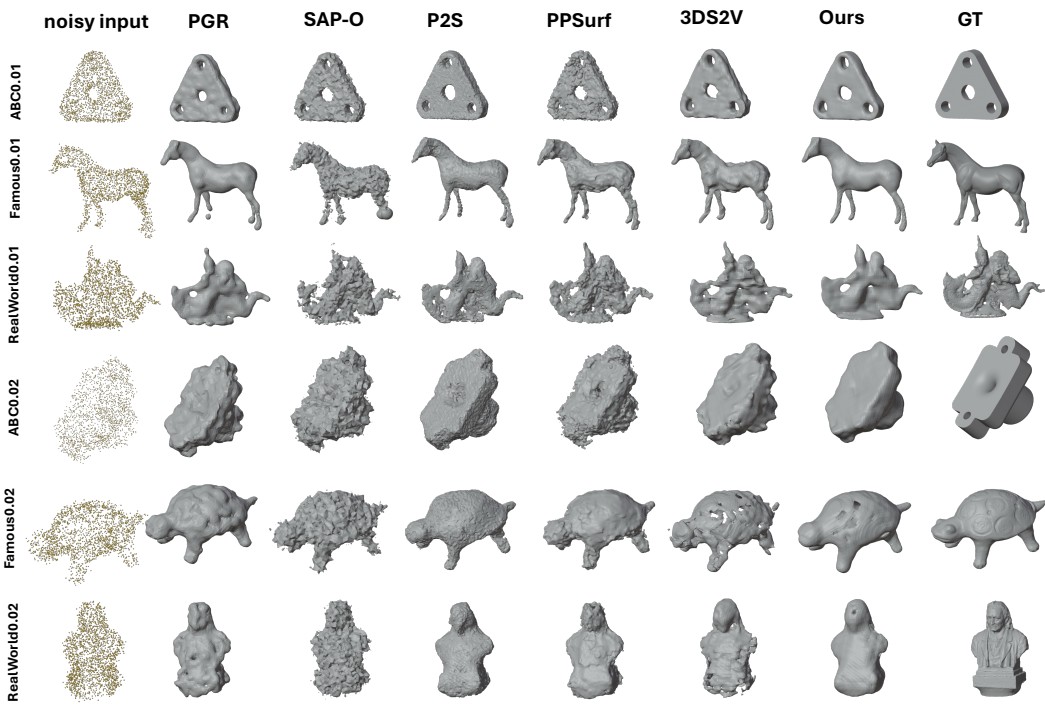

**Figure 4:** Visualization of surface reconstruction under Gaussian noise ($\sigma = 0.01$ and 0.02), comparing two overfitting-based methods (PGR [26] and SAP-O [36]), three data-driven approaches (P2S [14], PPSurf [15], 3DS2V [47]), and our proposed method. Results are shown for three benchmark datasets: ABC [22], Famous [14], and Real [14].

## 5.5 Ablation Study

Our ablation experiments were carried out on the "Chair" subset of ShapeNet, which contains 6271 models for training, 169 for validation, and 338 for testing. The configurations for all of our principal experiments originate with this subset and are progressively generalized to broader settings.

**Denoising Network.** The 3DS2V decoder contains 24 self-attention blocks, one cross-attention block, and a fully connected layer. Fine-tuning the entire decoder yields the best denoising performance, with metrics steadily improving over 15 epochs: IoU increases from 0.88 to 0.93, and Normal Consistency (NC) improves from 0.93 to 0.96. In comparison, fine-tuning only the fully connected layer offers virtually no benefit. Adding the cross-attention block introduces a clear gain—its final NC reaches 0.95, close to the best model—but at the cost of doubling the training time.

**Table 5:** Comparison of fine-tuning strategies on denoising performance.

| Metric | FC Layer Only | FC + Cross-Attention Block | Entire Decoder |
|--------|---------------|---------------------------|----------------|
| IoU | 0.88 (no gain) | 0.92 | **0.93** |
| NC | 0.93 (no gain) | 0.95 | **0.96** |
| Epochs | – | 30 | **15** |

**Noise Type.** Beyond standard zero-mean Gaussian noise, we evaluated three additional noise types—Uniform, Discrete, and Laplace noise—each applied at a fixed magnitude of $\sigma = 0.01$. Furthermore, to assess the impact of non-zero bias in Gaussian perturbations, we conducted experiments over the domain $[-1, 1]^3$ using means of $\mu = 0.005, 0.01$, and $0.02$. Comprehensive quantitative results are presented in Table 6.

At $\sigma = 0.01$ on $[-1, 1]^3$, our model consistently shows denoising performance under Uniform and Discrete noise, with notable gains in both IoU and NC. However, it shows limited benefit under Laplace noise. For Gaussian noise with $\sigma = 0.01$ increasing $\mu$, our model remains effective at lower $\mu$, but its advantage diminishes as the $\mu$ grows, eventually leading to degraded reconstruction quality.

**Table 6:** Performance comparison between the 3DS2V [47] and our method under various noise types and levels. Noise distributions tested are Uniform, Discrete and Laplace with $\sigma = 0.01$, and Gaussian with $\sigma = 0.01$ under different means $\mu = 0, 0.005, 0.01, 0.02$.

| Dataset | IoU ↑ | | NC ↑ | | Chamfer | | F-Score | |
|---------|-------|------|------|------|---------|------|---------|------|
| | 3DS2V | Ours | 3DS2V | Ours | 3DS2V | Ours | 3DS2V | Ours |
| Uniform ($\sigma = 0.01$) | 0.873 | **0.911** | 0.920 | **0.962** | 0.015 | **0.013** | 0.985 | **0.986** |
| Discrete ($\sigma = 0.01$) | 0.867 | **0.895** | 0.915 | **0.960** | 0.015 | **0.014** | 0.984 | **0.985** |
| Laplace ($\sigma = 0.01$) | 0.909 | 0.908 | 0.956 | **0.956** | 0.014 | **0.014** | 0.985 | **0.985** |
| Gaussian ($\sigma = 0.01$) | 0.887 | **0.927** | 0.937 | **0.966** | 0.014 | **0.013** | 0.986 | **0.986** |
| Gaussian ($\sigma = 0.01, \mu = 0.005$) | 0.860 | **0.881** | 0.942 | **0.964** | 0.017 | **0.016** | 0.982 | **0.984** |
| Gaussian ($\sigma = 0.01, \mu = 0.01$) | 0.798 | **0.800** | 0.946 | **0.949** | 0.024 | **0.023** | 0.952 | **0.960** |
| Gaussian ($\sigma = 0.01, \mu = 0.02$) | 0.661 | **0.666** | 0.875 | **0.898** | **0.038** | 0.039 | **0.549** | 0.524 |

## 6 Conclusion

In this work, we introduced NoiseSDF2NoiseSDF, a framework capable of recovering clean surfaces from noisy and sparse point clouds by leveraging paired noisy SDFs in a Noise2Noise denoising approach. We demonstrated that, at noise intensities of 0.01 and 0.02, the reconstructed surfaces produced by our method are notably cleaner and smoother, both quantitatively and visually, compared to previous works. In future research, we aim to explore additional applications of the NoiseSDF2NoiseSDF framework, such as scaling up point cloud sizes to enhance geometric detail recovery, or replacing components within the framework with alternative architectures trained from scratch, thus improving the model's ability to represent point cloud noise and further enhancing denoising performance.

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
