# OpenReview forum: "NoiseSDF2NoiseSDF: Learning Clean Neural Fields from Noisy Supervision"
_NeurIPS.cc/2025/Conference — Submitted to NeurIPS 2025_

### Official Review · Reviewer_HjJN · 2025-06-20

**Clarity:** 3
**Significance:** 4
**Originality:** 3
**Rating:** 5
**Confidence:** 5

**Summary:**

This paper introduces NoiseSDF2NoiseSDF, a novel method that extends the Noise2Noise paradigm to 3D neural implicit representations, specifically signed distance functions (SDFs). The core idea is to learn clean SDFs from noisy point cloud data by supervising with other independently noisy SDFs, without requiring access to clean ground-truth data. The method exploits the coordinate consistency of neural SDFs, enabling the use of simple MSE loss for denoising. Built on top of the 3DS2V framework, the approach shows significant improvements in surface reconstruction quality under varying levels of noise across multiple datasets.

**Questions:**

See weaknesses.

**Ethical Concerns:**

["NO or VERY MINOR ethics concerns only"]

**Limitations:**

yes

**Paper Formatting Concerns:**

I did not notice any major formatting issues. The paper appears to follow the NeurIPS 2025 formatting guidelines.

**Quality:**

3

**Strengths And Weaknesses:**

I find the Noise2Noise paradigm particularly appealing, and extending it from 2D images to 3D point clouds and further into SDF fields is a well-motivated and elegant direction. The adaptation to SDFs is intuitive and well-justified, leveraging their coordinate-aligned structure to enable effective supervision. While the network architecture and design choices are relatively simple and may not appear highly novel at first glance, I believe this is a valuable and meaningful attempt. The method is practical, requiring no clean labels, and remains agnostic to the specific Point2SDF backbone, making it broadly applicable. The overall pipeline is clearly described and thoughtfully implemented.
However, the method's reliance on pretrained Point2SDF networks (e.g., 3DS2V) as supervision raises questions about how much the final performance depends on the quality and noise sensitivity of those networks. Does the method still work when Point2SDF predictions are especially inconsistent? How sensitive is the denoising effect to architectural choices or training regimes of the supervising model?
While the paper compares with several baselines, it lacks a quantitative and visual comparison with Noise2Noise Mapping[1]—a key representative of Noise2Noise in the point cloud domain with open-source code available. A direct comparison, e.g., on a small dataset like Famous, would more clearly highlight the benefits of applying Noise2Noise in the SDF space over point-wise methods.

[1] Baorui Ma, Yu-Shen Liu, and Zhizhong Han. Learning signed distance functions from noisy 3d point clouds via noise to noise mapping. In Proceedings of the 40th International Conference on Machine Learning (ICML)

---

> ### Author Rebuttal · Authors · 2025-07-31
>
> We thank the reviewer for appreciating our well-motivated and elegant direction, the intuitive and well-justified adaptation, and the valuable and meaningful attempt. We also thank the reviewer for recognizing that our pipeline is clearly described and thoughtfully implemented.
>
> # Q1: Sensitivity to Supervising Model Design
>
> | Point2SDF | IoU ↑   | NC ↑    | CD-L2 ↓    | F-Score ↑ |
> |-----------|-------|-------|-------|---------|
> | 3DILG     | 0.913 | 0.962 | 0.014 | 0.978   |
> | 3DS2V     | 0.927 | 0.966 | 0.013 | 0.986   |
>
> We thank the reviewer for their question regarding the sensitivity of the Supervising Model design. To evaluate this, we conducted an ablation study using an alternative pretrained model, 3DILG, to generate the noisy supervision signals as Point2SDF.  3DILG encodes 3D shape using irregular latent grids, while 3DS2V encodes neural fields on top of a set of vectors. We then trained the denoising network with the same procedure and compared the results to those obtained using 3DS2V supervision. The results show that the final performance remains very close across both supervision sources, indicating that our method is not overly sensitive to the specific design or quality of the supervising model. This demonstrates that our framework is robust and can generalize well even when the supervisory signal comes from different pretrained SDF estimators.
>
> # Q2: Comparison with Noise2Noise Mapping
>
> | Method        | CD-L1 ↓ | CD-L2 ↓ | NC ↑    | F-Score ↑ |
> |:---------------:|:--------:|:--------:|:--------:|:---------:|
> | N2NM          | 0.026  | –      | 0.962  | 0.991   |
> | Ours          | 0.026  | 0.011  | **0.971**  | 0.985   |
>
> We thank the reviewer for pointing out the lack of comparison with Noise2Noise Mapping (N2NM). To address this, we conducted a comparison with N2NM across 13 categories of the ShapeNet dataset, following the exact evaluation setup and noise condition (Gaussian noise with standard deviation 0.005) used in the N2NM paper. In the above table, the results show that our method achieves better NC performance while maintaining a close F-Score and same CD-L1. Importantly, N2NM is an overfitting-based method that requires 46 minutes per shape for inference as shown in paper [10]. In contrast, our data-driven model generalizes across shapes, enabling fast inference in only 0.05 second. This demonstrates that our method offers a strong trade-off between quality and efficiency. We will add the comparison and discussion in our revised paper.
>
> On the Famous dataset, our method underperforms compared to N2NM, with higher CD-L2 for both medium noise (0.016 vs. 0.003) and maximum noise (0.032 vs. 0.012). This discrepancy may be attributed to N2NM using more input points and directly training on the Famous dataset, effectively overfitting to its shapes. In contrast, we did not perform any fine-tuning on Famous. Additionally, there exists a domain gap between our training data (ShapeNet) and the Famous dataset.

---

> ### Author Response · Authors · 2025-08-04
> **Q2 Famous results correction**
>
> After carefully reviewing the TPAMI version of N2NM [1], we found that the paper reports two types of CD metrics in Table IV for the Famous dataset. Upon further examination of the metric definitions and code implementations, we confirmed that our previously reported CD metric actually corresponds to the P2S-CD metric in [1]. To ensure a fair and accurate comparison with N2NM on the Famous dataset, we have updated our results accordingly. Moreover, we conducted test-time optimization (TTO) on the Famous dataset by fine-tuning our pretrained model with the NoiseSDF2NoiseSDF paradigm, achieving results comparable to N2NM under both noise conditions. The experimental data is summarized in the following table:
>
> | Method | Medium noise | Maximum noise |
> |--------|--------------|---------------|
> | N2NM   | 0.0132       | 0.0231        |
> | Ours   | 0.0160       | 0.0320        |
> | Ours-TTO| 0.0108      | 0.0252        |
>
>
> [1] Junsheng Zhou, Baorui Ma, Yu-Shen Liu, Zhizhong Han: Fast Learning of Signed Distance Functions From Noisy Point Clouds via Noise to Noise Mapping. IEEE Trans. Pattern Anal. Mach. Intell. 46(12): 8936-8953 (2024).

---

> > ### Comment · Reviewer_HjJN · 2025-08-08
> >
> > After reading the authors’ response and considering other reviewers’ comments, I tend to maintain my original score and recommend acceptance of the paper. I look forward to seeing the authors extend their experiments to broader scenarios in future work.

---

> > > ### Author Response · Authors · 2025-08-09
> > >
> > > We are grateful to the reviewer for the strong recognition of our work and the kind recommendation for acceptance. We also sincerely appreciate the valuable suggestion to extend our experiments to broader scenarios in our future work.

---

### Official Review · Reviewer_EMRx · 2025-06-26

**Clarity:** 3
**Significance:** 3
**Originality:** 3
**Rating:** 4
**Confidence:** 5

**Summary:**

This paper introduces a new approach for learning clean SDF from noisy point clouds, producing robust surface reconstruction. The key idea is to treat the 3D queries as the 2D pixels for a noise-to-noise learning. The results outperform several baselines, demonstrating the effectiveness of proposed method.

**Questions:**

Please refer to the weaknesses above. I am overall not negative with the proposed method, my main concerns are with the insufficient experiments. I am willing to raise my score if my concerns are well addressed.

**Ethical Concerns:**

["NO or VERY MINOR ethics concerns only"]

**Final Justification:**

Most of my concerns are adressed and I will raise my score to 4. The authors should add all the experiments during rebuttal to the paper for revision, especially the experiments in discussion with reviewer nTsM and EMRx.

**Limitations:**

Yes. In the appendix.

**Paper Formatting Concerns:**

The formatting of the figures and tables are not correct. The titles should not be centered.

**Quality:**

3

**Strengths And Weaknesses:**

Strengths
1. The paper is well written. Figure 1 clearly shows the key insight of the proposed method, making the idea easy to understand.
2. The idea is simple but works well. The insight is new which solves the difficulty in 3D point matching by learning with correspondences of 3D queries.

Weaknesses
1. The experiments should be improved. Some SOTA methods are not included in the comparison, which makes it hard to evaluate the performance of proposed method. For example, Neural IMLS, Noise2Noise Mapping, LocalN2NM. I understand that some of those methods are optimization-based, while a comparison is also recommended since all those methods are in the same setting.
2. What if we first clean the noisy point clouds with point denoising methods (e.g. IterativeFPN) and then reconstruct surfaces from the cleaned points? Will the proposed method still performs better?
3. The authors only show results under object-level models. Can the proposed method be used in reconstructing scene-level models with noisy scene point clouds as input?
4. The title of the figures and tables should not be centered, please check the format provided by the official.
5. Missing references to some new works. Fast Learning of Signed Distance Functions from Noisy Point Clouds via Noise to Noise Mapping [TPAMI24]. Noise4Denoise: Leveraging Noise for Unsupervised Point Cloud Denoising [CVM 24].

---

> ### Author Rebuttal · Authors · 2025-07-31
>
> We thank the reviewer for appreciating our method is new and simple but works well.
>
> # Q1: Comparison with optimization-based methods
> We thank the reviewer for suggesting the inclusion of optimization-based methods such as Neural-IMLS, Noise2Noise Mapping (N2NM), and LocalN2NM in our comparison. To address this, we have added results comparisons with Neural-IMLS, N2NM, and LocalN2NM on the 13 categories of ShapeNet dataset, under the same noise condition (Gaussian noise with standard deviation 0.005) following Neural-IMLS and N2NM. The results show that our method performs competitively with N2NM and LocalN2NM in terms of NC and F-Score, and outperforms Neural-IMLS by a significant margin. It is also worth noting that N2NM and LocalN2NM are overfitting-based methods that require long inference times per shape (46 minutes and 5 minutes, respectively) as shown in [10]. In contrast, our method is data-driven and generalizes well, allowing us to infer a single shape in just 0.05 second. These results demonstrate that our method strikes a strong balance between quality and efficiency. We will update our paper to include the comparative results and corresponding discussions.
>
> | Method        | CD-L1 ↓   | NC ↑    | F-Score ↑ |
> |:---------------:|:--------:|:--------:|:---------:|
> | Neural-IMLS   | 0.031  | 0.944  | 0.983   |
> | N2NM          | 0.026  | 0.962  | 0.991   |
> | LocalN2NM     | 0.023  | 0.973  | 0.992   |
> | Ours          | 0.026  | 0.971  | 0.985   |
>
>
> # Q2: Denoising Then Reconstruction
>
> | Method                     | IoU ↑  | NC ↑   | CD-L2 ↓   | F-Score ↑ |
> |----------------------------|--------|--------|--------|------------|
> | Denoising-Then-Reconstruction | 0.882 | 0.954 | 0.016 | 0.969   |
> | Ours                | **0.927** | **0.966** | **0.013** | **0.986**   |
>
> We thank the reviewer for this insightful question. Following this suggestion, we conducted an additional experiment: we first applied IterativeFPN to denoise the noisy point clouds and then used 3DS2V to reconstruct surfaces, using the Chair category from ShapeNet as the benchmark. We compared the resulting surfaces with those produced by our proposed method. The results show that our method performs better than this two-stage pipeline in terms of reconstruction quality.  This supports our key insight that directly learning from noisy data via neural SDFs with noisy supervision is not only feasible but also more effective than decoupling denoising and reconstruction. We will include these results in the revised version of our paper as well.
>
> # Q3: Reconstructing scene-level models with noisy scene point clouds as input?
>
> Thank you for raising this insightful and forward-looking question — we truly appreciate your perspective on the broader applicability of our method.
>
> While our current work focuses on object-centric datasets (e.g., ShapeNet-scale) due to resource and computational constraints, we agree that extending to larger and more complex scenes is an important direction for future work. Our method builds upon the 3DS2V framework, which was originally designed for object-centric datasets such as ShapeNet. Scaling this framework to large-scale or non object-centric datasets introduces several challenges: such datasets often involve more complex geometry and finer-grained spatial details, which require models to operate over larger and more diverse feature distributions. This would likely necessitate a higher-capacity network and potentially a different architectural design to preserve efficiency and reconstruction quality. Despite these challenges, we believe our noisy-supervision strategy remains applicable and promising in such settings. We plan to explore this extension in future work by incorporating more powerful backbone architectures capable of handling the increased complexity of large-scale or scene-level data.
>
> # Q4:  Format of the title of the figures and tables.
>
> Thank you for pointing this out. We appreciate the reminder and will make sure to revise all figure and table titles to follow the official formatting guidelines.
>
> # Q5: Missing references.
>
> Regarding *Fast Learning of Signed Distance Functions from Noisy Point Clouds via Noise to Noise Mapping* [TPAMI 2024], we would like to clarify that we have already cited the conference version of this work as citation [30]. We are happy to add the citation to the TPAMI version to reflect the most recent publication and latest research outputs. For *Noise4Denoise: Leveraging Noise for Unsupervised Point Cloud Denoising* [CVM 2024], we have already included this reference in our paper as citation [40]. We will discuss its relevance in the context of learning from noise in unsupervised settings. We hope this clears up the concern, and we appreciate the reviewer’s careful attention to related work.

---

> > ### Author Response · Authors · 2025-08-09
> >
> > Dear Reviewer,
> >
> > Following the initial review, we have made efforts with our best to address the raised issues. We would appreciate your feedback on whether our rebuttal has resolved all of your concerns. Your feedback is very important to us, and we are happy to provide any further clarification if needed. Thank you very much for your time and support.
> >
> > Sincerely,
> >
> > The Authors

---

### Official Review · Reviewer_EJ66 · 2025-07-02

**Clarity:** 2
**Significance:** 3
**Originality:** 3
**Rating:** 5
**Confidence:** 3

**Summary:**

This paper tackles the task of reconstructing implicit neural surfaces from noisy point cloud primitives by extending the Noise2Noise paradigm to Signed Distance Functions (SDFs). The entire framework operates under the mathematical premise that the observations of a given signal with an independent, zero-mean noise converge to the actual signal. The original Noise2Noise method demonstrated the effectiveness of utilizing noisy observations as supervisory signals with pixel-wise correspondence in the image denoising task. Instead of seeking point-wise correspondence brutally when moving from images to point clouds, the authors leverage the power of implicit neural surfaces to utilize the coordinates of SDF when building the correspondence required for the denoising procedure.

**Questions:**

1. See weaknesses.

2. While the authors include the source code of the proposed method in the supplementary material, I could not find either the Point2SDF part in the implementation or the exact method name of the used method. It would be better to elaborate on the architecture and method for completeness.

3. Regarding the experiment settings, is there any limitation on how each method can be evaluated on different datasets? If not, why is the 3DS2V method included in both Tables 1, 2, and 3 while some other methods are absent in some experimental settings?

**Ethical Concerns:**

["NO or VERY MINOR ethics concerns only"]

**Final Justification:**

I'd like to maintain my initial rating for this paper.
This paper is technically sound and provides valuable application potential in 3D reconstruction where noise in inevitable and sampling density is limited. Most of my concerns have been adequately addressed and the presented experimental results are convincing. While the proposed method lacks experiments in real-world objects and large scenes, after reading the comments from both peer reviewers and AC, I believe this method can serve as a solid foundation and provide useful insights for followers to explore such directions.

**Limitations:**

Yes.

**Paper Formatting Concerns:**

None.

**Quality:**

3

**Strengths And Weaknesses:**

**Strengths**

1. The proposed method offers a novel insight into surface reconstruction from noisy point clouds, which has potential applications in various fields.

2. The idea of leveraging coordinate correspondence in SDF as a workaround for finding point-wise correspondence directly between two given noisy point clouds is innovative and technically sound.

3. According to the experimental results, the proposed method has noticeable improvements over the existing methods in both quantitative and qualitative evaluations.

**Weaknesses**
1. While the proposed method demonstrates quantitative improvements over previous approaches, the qualitative results (e.g., Figure 4) indicate that this method struggles to reconstruct shapes with high-frequency details and thin structures.

2. The authors claim that the proposed method does not impose limitations on the architecture of SDF prediction network (the Point2SDF part). The manuscript itself does not include any related ablation study to support this generalization ability.

3. The authors utilize the encoder and decoder structure proposed by 3DS2V, and also include 3DS2V in their comparison, and in my understanding, the comparison is made using the shape autoencoding pipeline from 3DS2V. The proposed method has two noisy point cloud inputs, while the 3DS2V pipeline only sees one noisy point cloud. How to keep the comparison fair when input information is not balanced between different methods.

---

> ### Author Rebuttal · Authors · 2025-07-31
>
> We thank the reviewer for appreciating our novel insights, innovative and technically sound idea, and the noticeable improvements demonstrated in the experimental results.
>
> # Q1: The qualitative results (e.g., Figure 4)
>
> We thank the reviewer’s concern regarding the qualitative results. Reconstructing from sparse and noisy point clouds presents inherent challenges. Sparsity limits the amount of recoverable geometric information, particularly high-frequency details, while strong noise can destabilize the reconstructed geometry, often causing thin structures to break. Denoising methods face a trade-off between effective noise removal and over-smoothing. As shown in Figure 4, all baseline methods struggle to strike a balance between these two issues in their reconstructions. Although our method also has difficulty recovering fine details and thin structures, it performs better at preserving the main structure and overall geometry of the object, such as large and smooth surfaces. These features correspond to low-frequency components. In this respect, our method outperforms the baselines by producing cleaner, more coherent surfaces with fewer artifacts and fewer broken regions.
>
> # Q2: The generalization on the architecture of SDF prediction network (the Point2SDF part).
>
> | Point2SDF | IoU ↑   | NC ↑   | CD-L2 ↓    | F-Score ↑ |
> |:-----------:|:-------:|:-------:|:-------:|:---------:|
> | 3DILG     | 0.913 | 0.962 | 0.014 | 0.978   |
> | 3DS2V     | 0.927 | 0.966 | 0.013 | 0.986   |
>
> We appreciate the reviewer’s question concerning the generalization ability of Point2SDF. To validate that our framework does not impose constraints on the architecture of the SDF prediction network (i.e., the Point2SDF component), we conducted an additional ablation study on the Chair subset of ShapeNet by replacing the original 3DS2V module with 3DILG. 3DILG encodes 3D shape using irregular latent grids, while 3DS2V encodes neural fields on top of a set of vectors. The results show that our method remains effective in learning clean SDFs from noisy supervision. These findings support the generalizability of our approach across different SDF predictions. We will include this ablation study in the revised version of the paper to strengthen our claim.
>
> # Q3: Input information balance between different methods.
>
> To clarify, our method uses two independently corrupted point clouds only during training. One is passed through the Point2SDF module to generate a noisy supervision signal, while the other serves as the input to the Denoising Network, which is optimized using an MSE loss between its prediction and the noisy SDF.  Importantly, we use exactly the same dataset and training splits as 3DS2V, without introducing any additional data. The key difference is that our method adopts a noise2noise supervision scheme to demonstrate the effectiveness of the Noise2Noise framework in denoising SDFs.
>
> During evaluation, our method uses only a single noisy point cloud as input, which is exactly the same as in the original 3DS2V pipeline. No additional information is available to our model at inference time. This design ensures that all methods are using the same traing data and evaluated under identical testing conditions, making the comparison both fair and consistent.
>
> # Q4: Point2SDF part in the implementation.
>
> The main architecture and training code of our method can be found in: support/code/models/noise2noisefields.py. The Point2Sdf module is loaded at line 19 and is called fix_model. We will clarify in the updated code. To address the reviewer’s concern, we will rename and document the module more clearly in the camera-ready version to make the role of Point2Sdf and other components of our pipeline explicit and easy to follow.
>
> Thank you very much for your careful review and helpful comments on our code. We will further refine the implementation and make the codebase publicly available shortly after the NeurIPS review period.
>
> # Q5: Regarding the experiment settings and evaluation on different datasets.
>
> We appreciate the reviewer’s concern regarding the experiment settings. Our evaluation design is guided by principles of fairness, relevance, and computational feasibility, with each table addressing a specific aspect of the study.
> Tables 1, 2, and 3 primarily focus on comparisons with data-driven methods. Specifically, Tables 1 and 2 evaluate denoising performance on ShapeNet under varying noise levels. We include 3DS2V in these tables to highlight the direct improvements achieved by our method under consistent experimental conditions (i.e., identical dataset splits, noise levels, and evaluation metrics). Table 3 provides a broader comparison among data-driven approaches on ABC, Famous, and Real. As 3DS2V is also a data-driven baseline, its inclusion allows readers to better contextualize our method's performance within this category. Data-driven methods such as Points2Surf (P2S) and PPSurf require watertight meshes for training. However, as acknowledged by the authors of those works, ShapeNet is generally non-watertight, which limits compatibility with their training pipelines. Adapting or retraining these methods on ShapeNet to match our experimental setup would have required substantial computational resources beyond our current scope. For this reason, they are not included in Tables 1 and 2. In Table 4, we focus on comparisons with overfitting-based methods.  We will clarify this in the final version of the paper.

---

> > ### Comment · Reviewer_EJ66 · 2025-08-07
> >
> > I'd like to thank the authors for their elaborate response. My questions and concerns have been addressed accordingly. And thus I will retain my original review and recommend that this paper be accepted.
> > While the idea behind this proposed method is generalizable to arbitrary point clouds, the experiment in the manuscript focused on the synthetic objects. I suggest that the authors extend their experiments to different scenarios, including large scenes, the human body, and real-world targets.

---

> > > ### Author Response · Authors · 2025-08-08
> > >
> > > We gratefully thank the reviewer for the thoughtful feedback and kind recommendation for acceptance. We appreciate the valuable suggestion to extend our experiments to broader scenarios, and we will actively pursue this as an important direction in our future work.

---

### Official Review · Reviewer_nTsM · 2025-07-06

**Clarity:** 3
**Significance:** 2
**Originality:** 3
**Rating:** 4
**Confidence:** 4

**Summary:**

The paper tackles the problem of implicit surface reconstruction from sparse noisy point clouds. It proposes a method to finetune the decoder of a generalizable model (point-to-SDF model) by minimizing the squared difference between the sdf fields obtained by conditioning on two different noisy version of the same input point cloud. The effectiveness of method is evaluated by training and testing on the ShapeNet dataset.  The generalization capability of the method is assessed by training on the train set of the ABC dataset and evaluating on the Famous and Reak datasets.

**Questions:**

- I didn't understand what the authors mean by "To ensure that all SDF targets are consistently noisy, we freeze its parameters during this process". Does it mean  the pretrained 3DS2V model is fixed  during the whole training. Or you just detach the SDF prediction corresponding to p1 from the computational graph ?

**Ethical Concerns:**

["NO or VERY MINOR ethics concerns only"]

**Final Justification:**

The paper proposes a finetuning procedure to improve the generalization of Point2SDF backbones using a Noise2Noise framework. Comparisons on **Out of distribution datasets (ABC, Famous)** show that the proposed method is competitive with other alternative finetuning procedures (Meta-Learning,Closed form decoders (NKSR, Neural IMLS), test time optimization, single scene optimization Noise2Noise Mapping(N2NM) ).  While it doesn't outperform Noise2Noise Mapping(N2NM)  is all setting, the authors show that besides being significantly faster that N2NM at inference, the proposed method can benefit from the learned prior through test time finetuning and is able to recover thin structures of challenging shapes under large noise perturbations.

Overall, I think this is good contribution and l'm leaning towards acceptance.

**Limitations:**

yes

**Paper Formatting Concerns:**

No major formatting issues

**Quality:**

3

**Strengths And Weaknesses:**

## Strengths
The paper is well written and easy to follow.
- The proposed method is clearly presented and motivated as a rigorous application of the  *Noise2Noise* approach to Signed distance functions.
- The proposed method outperforms it’s baseline across datasets and metrics.

## Weaknesses
- **Missing Comparaison and discussion:**
    - w.r.t generalizable models NKSR, NoKSR, FS-SDF
    - w.r.t to recent single shape optimisation methods especially those dedicated to sparse and noisy point clouds  (NTPS, SparseOcc, Noise2Noise Mapping [30])*
    - w.r.t Noise2Noise Mapping [30]: I think comparaison to this baseline is important as it introduces the noise2noise paradigm for SDF learning and shows very competitive results to generalizable methods.
- **Motivation and ablations:**
    1. Why using   noisy-target supervision in neural SDFs for surface reconstruction is good idea ? I thing to answer this question some ablations are needed*
        1. The proposed loss  function can be seen as encouraging the model the be invariant to the noise on the input point cloud.  How does the method compare to direct sdf supervision  using the same batch configuration (same batch size , same number of shapes per batch, etc)
        2. Why is the loss used to finetune the decoder but not the encoder: Instead of finetuning the decoder such as that it produces the same sdf using latent features from different noisy versions of the same input why not simply apply the noise2noise loss to the encoder’s output + direct sdf supervision of the decoder ?
- Experiments on non object centric datasets/ large scale datasets (Faust , Scannet, Synthetic Rooms) would strengthen the paper.

---

> ### Author Rebuttal · Authors · 2025-07-31
>
> We thank the reviewer for appreciating that our method is clearly presented and motivated.
>
> # Q1: Comparison and Discussion
>
> We thank the reviewer for raising the point regarding comparisons with generalizable surface reconstruction models, NKSR, NoKSR, and FS-SDF. In response, we conducted experiments on the ShapeNet dataset following NKSR, using the same noise setting (Gaussian noise with a standard deviation of 0.005), and reported performance in terms of CD-L1, CD-L2, and IoU. The results are summarized below:
>
> | Method   | CD-L1 ↓  | CD-L2 ↓	  | IoU ↑  |
> |:----------:|:--------:|:--------:|:--------:|
> | NKSR     | 0.0234 | –      | 95.60  |
> | NoKSR    | 0.0288 | –      | 94.60  |
> | FS-SDF   | –      | 0.0290 | 87.00  |
> | Ours | 0.0260 | 0.0150 | **96.24**  |
>
> Our method achieves higher IoU (96.24\%) than all other methods. This indicates that our model is able to learn stronger representations from sparser and noisier input data, highlighting its data efficiency.
> While NKSR achieves a slightly lower CD-L1 (0.0234 vs. our 0.0260), our method achieves a significantly lower CD-L2 (0.0150 vs. 0.0290 for FS-SDF). CD-L2 is particularly relevant in noisy reconstruction, as it reflects the model's ability to avoid large geometric outliers.
>
> To compare with sparse point clouds methods (NTPS, SparseOcc), we conducted experiments following the setup used in SparseOcc, evaluating on the Chairs, Tables, and Lamps categories from the ShapeNet under Gaussian noise with standard deviation 0.005. The comparison is summarized below:
>
> | Method     | CD-L1 ↓  | CD-L2 ↓  | NC  ↑   | F-Score ↑ |
> |:------------:|:--------:|:--------:|:-------:|:---------:|
> | NTPS       | 0.111  | 0.067  | 0.880 | 0.740   |
> | SparseOcc  | 0.076  | 0.020  | 0.880 | 0.830   |
> | Ours   | **0.025**  | **0.011**  | **0.975** | **0.990**   |
>
> Our method outperforms both NTPS and SparseOcc by large margins across all metrics.  Our method demonstrates that leveraging this information with a data-driven network yields far better results than single shape optimization-based techniques which struggle to regularize under sparsity.
>
> To compare with Noise2Noise Mapping(N2NM), We conducted experiments following the same protocol and dataset split as the N2NM paper, evaluating on 13 categories of ShapeNet under Gaussian noise with standard deviation 0.005. The comparison is shown below:
>
> | Method        | CD-L1 ↓ | CD-L2 ↓ | NC ↑   | F-Score ↑ |
> |:---------------:|:--------:|:--------:|:--------:|:---------:|
> | N2NM          | 0.026  | –      | 0.962  | 0.991   |
> | Ours          | 0.026  | 0.011  | **0.971**  | 0.985   |
>
> Our method achieves a higher Normal Consistency (NC) while maintaining a competitive F-Score and matching the CD-L1 performance of N2NM. A limitation of N2NM lies in its overfitting-based design, which requires approximately 46 minutes of inference per shape as shown in [10]. In contrast, our data-driven approach generalizes across shapes and completes inference in just 0.05 second per shape. While N2NM performs optimization on a single shape in each run, our model is trained to denoise across an entire dataset, enabling it to generalize effectively to unseen noisy shapes without requiring per-shape re-optimization. To provide a more comprehensive comparison, we will include additional discussions and results involving the recommended methods in the revised version of the paper.
>
> # Q2: Why using noisy-target supervision in neural SDFs for surface reconstruction is a good idea?
>
> | Method                | IoU ↑   | CD-L2 ↓     | F-Score ↑ | NC ↑     |
> |:-----------------------:|:-------:|:--------:|:-------:|:--------:|
> | 3DS2V              | 0.887 | 0.014 | 0.986 | 0.937 |
> | Clear Supervision     | 0.939 | 0.013 | 0.988 | 0.970 |
> | Our Noisy Supervision | 0.927 | 0.013 | 0.986 | 0.966 |
>
> We appreciate the reviewer’s insightful question regarding the comparison to direct SDF supervision using the same batch configuration. Direct SDF supervision corresponds to the Noise2Clean paradigm in the context of Noise2Noise literature, where clean targets are used during denoising training. In our work, we simulate a more challenging scenario: learning clean neural SDFs from noisy point clouds using noisy supervision, analogous to Noise2Noise. We included comparisons under identical training conditions (same batch size, number of shapes per batch, etc.) between our proposed noisy-supervision method and direct SDF supervision (i.e., clean supervision) on the Chair subset of ShapeNet with noise level 0.01. These results are shown in table. We observe that our noisy supervision approach achieves performance nearly equivalent to that of clean supervision. This confirms our central hypothesis: it is possible to learn to produce cleaner outputs simply by observing noisy neural fields.
>
> # Q3: Why is the loss used to finetune the decoder but not the encoder?
>
> Our primary objective in this work is to investigate a more challenging and practical learning scenario: whether it is possible to learn clean neural SDFs directly from noisy point clouds using noisy supervision, without relying on noisy-clean pair targets. This aligns with the Noise2Noise paradigm, where supervision comes from noisy observations. While applying Noise2Noise loss at the encoder level along with direct SDF supervision at the decoder could be a potentially beneficial direction, it introduces clean supervision into the pipeline and becomes a noisy-clean training setup. We agree that this could be considered a strong baseline or even upper bound, but such strategies are beyond the primary scope of this paper, which is to explore noisy supervision as a learning signal. We performed an ablation study on the Chair category by progressively increasing the number of fine-tuned layers, starting from the decoder alone to the full encoder-decoder network. The results indicate that fine-tuning the entire encoder-decoder (IoU: 0.92, NC: 0.96) does not provide performance improvements over fine-tuning only the decoder (IoU: 0.93, NC: 0.96), while incurring longer training time and higher GPU memory consumption.
>
> # Q4: Experiments on non object-centric datasets / large-scale datasets
>
> While our current work focuses on object-centric datasets (e.g., ShapeNet-scale) due to resource and computational constraints, we agree that extending to larger and more complex scenes is an important direction for future work. Our method builds upon the 3DS2V framework, which was originally designed for object-centric datasets such as ShapeNet. Scaling this framework to large-scale or non object-centric datasets introduces several challenges: such datasets often involve more complex geometry and finer-grained spatial details, which require models to operate over larger and more diverse feature distributions. This would likely necessitate a higher-capacity network and potentially a different architectural design to preserve efficiency and reconstruction quality. Despite these challenges, we believe our noisy-supervision strategy remains applicable and promising in such settings. We plan to explore this extension in future work by incorporating more powerful backbone architectures capable of handling the increased complexity of large-scale or scene-level data.
>
> # Q5: Clarification on "freezing parameters'' during noisy SDF target generation
>
> To clarify, in our implementation, we load two instances of the 3DS2V model into GPU memory. One is completely frozen throughout training, which corresponds to the *Point2SDF* module shown in `Figure 2`. Its sole function is to provide a noisy SDF prediction based on input point cloud $p_2$, which serves as the noisy supervision signal. Parameter updates occur only in the other 3DS2V model, which we refer to as the *Denoising Network*. This setup ensures that the target SDF remains consistently noisy and unaffected by the optimization process, aligning with the core Noise2Noise training paradigm. We apologize for any confusion caused and will revise the related descriptions in `Sec. 4.2 Implementation` accordingly in the final version.

---

> ### Comment · Reviewer_nTsM · 2025-08-02
>
> I would like to thank the authors for their detailed explanations. While I appreciate the added experiments  the motivation of the proposed Noise2Noise method is not yet clear to me.
>
> - My understanding is that this framework could be interesting to improve the generalization of Point2SDF models to Out of Distribution datasets where only a set of scans (without GT SDF)  is available as a finetuning dataset. Finetuning on synthetic datasets where GT SDF is available seems contre-intuitive (which is confimed by the comparison to Clean Supervision). For this reason, I think comparisons to other methods especially single scene optimisation ones should be  performed on  out of distribution datasets. These methods (at least NKSR and POCO ; see Table 1 in NKSR) should figure in table 4 in order for the reader to get a full picture of the generalization ability of the proposed Noise2Noise finetuning method compared to these baselines.
>
> - The comparison to Noise2Noise Mapping(N2NM) seems to  limits the contribution of the paper to being a fast version of N2NM. Can the proposed method benefit from the learned prior at test-time: One way to show an advantage wrt N2NM is to do some finetuning steps at test time given the input pointcloud using the Noise2Noise loss. Does it improve the performance ?
>
> - Since the contribution of the paper is about the learning paradigm I think the comparison to FS-SDF can be improved. This paper (FS-SDF)  seems to use a very old backbone (IF-Net). As a consequence, it's not clear if the performance gap is due to the quality of the backbone or the finetuning strategy (Noise2Noise finetuning  vs Metalearning (learning to finetune in a few steps  with SDF =0 at the input pointcloud). In general, if the paper claims to address the generalization problem, I would appreciate a larger discussion about the different ways (Meta-Learning,Closed form decoders (NKSR, Neural IMLS), test time optimization, single scene optimization, etc )  to tackle this problem and how the paper positions itself wrt to these methods both theoretically and in terms of performance.
>
> Overall, I believe this is a good contribution but I think the motivation should be clarified and the experiments and comparisons shoud be made compatible with it. I'm willing to raise my rating if these points can be addressed.

---

> ### Author Response · Authors · 2025-08-05
>
> # Generalization to Out of Distribution datasets
>
> We thank the reviewer for the insightful observation regarding the applicability of our framework to out-of-distribution (OOD) datasets where only raw scans (without ground-truth SDFs) are available. We agree that fine-tuning on synthetic datasets with access to clean supervision may seem less intuitive in this context. Our intent was not to claim superiority over clean supervision, but rather to validate the Noise2Noise paradigm in a controlled setting. These experiments demonstrate that learning from noisy SDFs alone can achieve performance comparable to models trained with clean supervision, consistent with the original findings of the Noise2Noise framework in the 2D image domain. Moreover, in our ABC-related experiments, we apply the Point2SDF model pretrained on ShapeNet, and finetune this pretrained model on the noisy ABC train set using our NoiseSDF2NoiseSDF paradigm without relying on any clean ground-truth data from the ABC dataset. The experimental results in Tables 3 and 4 of our paper can demonstrate our paradigm’s ability to improve the generalization of Point2SDF to OOD datasets by only finetuning on noisy data.
>
> We appreciate the reviewer’s insightful suggestion. In response, we have followed similar evaluations used in Table 1 of the NKSR paper, and compared with NKSR and POCO for a more comprehensive evaluation of the generalization ability of our Noise2Noise fine-tuning method on out-of-distribution (OOD) datasets. We used the official pretrained models for both methods. Since NKSR requires point normals as additional input for surface reconstruction, we computed them using open3D python library. In the following table, we present the results under noise level 0.01, which demonstrate that our method achieves competitive performance compared to these strong baselines, despite not relying on clean SDF supervision. We will update our final paper with complete results accordingly.
>
> | Dataset    | **CD-L2 ↓** |        |        | **F-Score ↑** |        |       |
> | -----------| ------------- | ------ | ------ | ------------- | ------ | ----- |
> |            | POCO   | NKSR   | Ours      | POCO      | NKSR   | Ours  |
> | ABC    | 0.014  | 0.018  | **0.014** | **0.941** | 0.927 | 0.938 |
> | Famous | 0.017  | 0.017  | **0.016** | 0.923     | **0.943** | 0.941 |
> | Real   | 0.017  | 0.015  | **0.015** | 0.927     | 0.955 | **0.956** |
> | mean   | 0.016  | 0.017  | **0.015** | 0.930     | 0.941 | **0.945** |
>
> **Note**: Based on our recent experiments, we observed that the performance of the NKSR method is highly sensitive to the quality of input normals. This observation suggests that the generalization ability and practical applicability of our method are superior to NKSR, as high-quality normals are often difficult to obtain from raw point clouds or LiDAR scans in real-world scenarios.
>
> # Test-Time Optimization
>
> We thank the reviewer for raising this important point regarding the distinction between our method and Noise2Noise Mapping (N2NM). In response to the reviewer’s suggestion, we conducted experiments incorporating test-time optimization (TTO) using our Noise2Noise loss on ShapeNet. We found that TTO consistently improved our method’s performance, and the resulting Ours-TTO variant achieves even better results than N2NM. We will include these TTO results in the revised version.
>
> | **Method** | **CD-L1 ↓** | **CD-L2 ↓** | **NC ↑** | **F-Score ↑** |
> |------------|-------------|-------------|----------|----------------|
> | N2NM       | 0.026       | –           | 0.962    | 0.991          |
> | Ours       | 0.026       | 0.011       | 0.971    | 0.985          |
> | Ours-TTO   | 0.025       | 0.010       | 0.975    | 0.989          |
>
> Moreover, we conducted TTO experiments on the Famous dataset. Our method, when combined with TTO, not only demonstrated significant self-improvement, but also outperformed N2NM under medium noise conditions and achieved comparable results under maximum noise. The experimental results are summarized in the table below.
>
> | Method | Medium noise (CD-L2) | Maximum noise (CD-L2)|
> |--------|--------------|---------------|
> | N2NM   | 0.0132       | 0.0231        |
> | Ours   | 0.0160       | 0.0320        |
> | Ours-TTO| 0.0108      | 0.0252        |
>
> Overall, our method shows improved performance when fine-tuning is performed at test time on the input point cloud using the Noise2Noise loss. We sincerely appreciate the reviewer’s insightful suggestion, which has helped to further highlight the potential of our method from a broader perspective.

---

> ### Author Response · Authors · 2025-08-05
>
> # Metalearning  Comparison
>
> Thank you for the insightful suggestion. To more clearly attribute performance differences to the learning paradigm, we followed the reviewer's suggestion and re-implemented the metalearning finetuning using the same 3DS2V backbone as our method. Specifically, we trained the model to finetune in a few steps with SDF = 0 at the input point cloud. We conduct the experiments on the Chair of ShapeNet with noise level 0.01. While the usage of stronger backbone led to improved performance for the meta-learning method, the results still fall short compared to our Noise2Noise finetuning strategy.
>
> | **Method**     | **IoU ↑** | **CD-L2 ↓** | **F-Score ↑** | **NC ↑** |
> |----------------|-----------|-------------|----------------|----------|
> | 3DS2V          | 0.887     | 0.014       | 0.986          | 0.937    |
> | 3DS2V+Metalearning  | 0.901     | 0.014       | 0.986         | 0.941    |
> | 3DS2V+Noise2Noise (Ours) | 0.927     | 0.013       | 0.986          | 0.966    |
>
> # Larger discussion
>
> We appreciate the reviewer’s thoughtful suggestion regarding a more larger discussion on generalization paradigms. We will revise the manuscript to include a dedicated discussion in the Related Work (Section 2) that covers the following:
> - Meta-learning approaches (e.g., FS-SDF) treat the reconstruction task as a few-shot problem, training the network such that it can adapt to a new shape in a few gradient steps. These methods require test-time fine-tuning to adapt to new inputs.
> -  Closed-form methods like NKSR represent surfaces using compactly supported kernel functions and reconstruct shapes by solving a linear system that fits both point positions and normals. Similarly, Neural-IMLS employs moving least-squares (MLS) interpolation to define locally implicit surface functions from point clouds.
> - Test-Time Optimization (TTO) refers to the process of adapting a pretrained model to each specific input instance during inference. TTO typically fine-tunes the model using the unseen input point cloud at test time, which offers improved accuracy but introduces additional computational cost during inference.
> - Overfitting-based methods (e.g., DeepSDF, N2NM) train one model per shape. These often offer high fidelity but lack scalability and generalization, requiring significant computation per scene.
> - Our method builds on 3DS2V and is trained using a Noise2Noise strategy by minimizing MSE between noisy SDFs to recover clean implicit surfaces directly from noisy point clouds. This approach uniquely allows us to:
> (i) train with noisy supervision;
> (ii) generalize to unseen shapes without fine-tuning;
> and (iii) support test-time optimization for further performance improvement.
> If you find any revisions inappropriate or notice any missing discussions, we would greatly appreciate your feedback and will update the manuscript accordingly. Thank you again for your valuable comments!

---

> ### Comment · Reviewer_nTsM · 2025-08-06
>
> Most of my concerns have been addressed. Consequently I decided to increase my score. However, I still think this is borderline paper.  The comparison to N2NM is my  major concern as it seems to be more robust to noise and obtains similar results without relying on a prior. Please clarify this point. More precisely,
>
> - **Is the proposed method just a fast N2NM ?**
> - Can you show any other benefits of using a prior and initialisation of the decoder in this case ?
> - Are there case where N2NM while fail while the proposed method perfoms well? For example thin structures in classes like lamp or similar ones in Thingi10k or Famous. At high levels or noise N2NM should not be able to recover these structures, while a good feature representation may help ?

---

> > ### Author Response · Authors · 2025-08-08
> >
> > We are pleased to see that our responses have addressed most of the reviewer’s concerns. We would like to thank the reviewer for an increased rating of our work. To further address the major concerns regarding the comparisons to N2NM, we would like to provide the following responses:
> >
> > # Is the proposed method just a fast N2NM ?
> >
> > While both approaches are inspired by the Noise2Noise philosophy--which motivates the exploration of 3D reconstruction with noisy supervision--they are fundamentally different in design, representation, and usability.
> >
> > - N2NM is an overfitting-based method that performs denoising directly in the unstructured **point cloud domain** for each individual object. In this domain, there is **no explicit point-to-point correspondence** between different noisy point clouds, as shown in Figure 1(b) of our paper. To address this, N2NM devises the EMD loss to establish only **soft correspondences** between point sets. This solution introduces computational overhead and inherent approximation errors. Additionally, N2NM must be trained from scratch for every single shape, resulting in long inference times, limited scalability, and no ability to generalize across different shapes. In practice, N2NM requires a large number of noisy point clouds for accurate inference, e.g., 200 samples were used in their experiments.
> >
> > - In contrast, our method is a generalizable, data-driven approach that operates in the **neural fields domain** like SDFs. In this domain, **coordinate-to-coordinate correspondence** is inherently preserved between two noisy neural fields representing the same shape, as illustrated in Figure 1(c) of our paper. This coordinate alignment allows us to use a simple and effective MSE loss, similar as the original Noise2Noise where pixel-wise MSE ensures every pixel in the noisy samples corresponds to the same underlying signal. Thus, our method avoids the need for parameter-sensitive losses or complex point set association strategies required by N2NM, resulting in more consistent and computationally efficient performance. Once trained, our model can reconstruct any previously unseen shape using only a single noisy point cloud as input, without any need for per-shape optimization.
> >
> > # Other benefits
> >
> > - As previously explained, our method does not require training on the entire dataset; rather, it only needs to be fine-tuned on OOD datasets to achieve strong performance. This advantage is largely due to our use of a learned prior, such as 3DS2V, which effectively captures generalizable features from large-scale data. By leveraging this prior, our method can rapidly adapt to new data distributions with minimal fine-tuning, significantly reducing the computational and data requirements typically needed for retraining, as indicated in our comparisons to other data-driven approaches.
> > - By incorporating a learned prior, our approach achieves an architecture that is not restricted to a specific 3D feature representation. This means that the learned prior can be flexibly replaced or extended to support other 3D alternatives as needed in different datasets or tasks. With strong extensibility and practical value, our method showcases how large pretrained models can be effectively leveraged to address a variety of 3D vision challenges using limited resources.

---

> > > ### Author Response · Authors · 2025-08-08
> > >
> > > # Thin structures experiments
> > >
> > > We thank the reviewer's insightful comments that well align with our experimental observations. Compared to N2NM, our method demonstrates superior performance in cases involving thin structures and when there is large noise. For example, in the lamp category of ShapeNet, where samples often contain thin structural components, our method not only outperforms N2NM across all evaluation metrics but also produces higher-quality reconstruction results—especially in regions with long, narrow lamp stands and delicate connecting arms. This improvement can be attributed to the integration of global latent feature priors learned by 3DS2V, coupled with our Noise2Noise strategy operating in the SDF domain. Together, our method manages to recover fine-grained geometric structures and significantly enhance reconstruction quality.
> > >
> > > | Method | CD-L1 ↓ | NC ↑  | F-Score ↑ |
> > > |--------|---------|-------|------------|
> > > | N2NM   | 0.027   | 0.957 | 0.990      |
> > > | Ours   | 0.022   | 0.976 | 0.992      |
> > >
> > > We also evaluate the performance of N2NM and our proposed method on the Hand from the Famous dataset and Moon Dragon Castle (MDC) from Thingi10K under three levels of noise: mild (0.01), medium (0.02), and high (0.03). These selected objects feature elongated and geometrically challenging parts, including articulated, joint-based structure, as well as sparsely connected elements like claw tips. The results show that our method consistently outperforms N2NM across all noise levels. Notably, the performance gap widens as noise increases, which clearly demonstrates the robustness of our approach under challenging conditions. For the Hand object, as the noise level increases, N2NM struggles to separate the fingers, whereas our method successfully preserves the main finger structure. For Moon Dragon Castle, N2NM fails to capture the overall structure of the claws, while our method maintains the claw geometry with smooth surfaces. These results demonstrate the robustness of our method under high levels of noise. We will include these visual comparisons in the supplementary material.
> > >
> > > | Hand | Method | CD-L2 ↓   | F1 ↑         |
> > > |-------|--------|----------|--------------|
> > > | Mild  | N2NM   | 0.0123   | 0.988       |
> > > |       | Ours   | 0.0122   | 0.990       |
> > > | Medium| N2NM   | 0.0540   | 0.578       |
> > > |       | Ours   | 0.0229   | 0.837       |
> > > | High  | N2NM   | 0.0525   | 0.512       |
> > > |       | Ours   | 0.0284   | 0.757       |
> > >
> > > | MDC | Method | CD-L2 ↓   | F1 ↑   |
> > > |--------|--------|--------|--------|
> > > | Mild   | N2NM   | 0.0148 | 0.978 |
> > > |        | Ours   | 0.0104 | 0.997 |
> > > | Medium | N2NM   | 0.0292 | 0.744 |
> > > |        | Ours   | 0.0212 | 0.886 |
> > > | High   | N2NM   | 0.0407 | 0.547 |
> > > |        | Ours   | 0.0274 | 0.778 |

---

### Note · Authors · 2025-08-15

Dear ACs and Reviewers,

As the first work that extends the Noise2Noise paradigm to the neural fields, our NoiseSDF2NoiseSDF method learns clean neural SDFs from noisy supervision by leveraging inherently maintained coordinate-to-coordinate correspondence, achieving good performance with fast inference speed, as well as robustness and strong generalization ability. We are pleased to see the following highlights that have been commonly recognized by the reviewers:

- **Novelty and value of the insight** – The paper’s insight is novel (EJ66) and new (EMRx), representing a valuable and meaningful attempt (HjJN). The idea is innovative (EJ66), intuitive and well-justified (HjJN), with the extension of Noise2Noise to SDF fields being well-motivated (nTsM, HjJN) and an elegant direction (HjJN).
- **Technical soundness and practicality** – The work is technically sound (EJ66), with a pipeline that is clearly described and thoughtfully implemented (HjJN), and the method is practical (HjJN). It is simple but works well (EMRx).
- **Clarity of presentation** – The method is clearly presented (nTsM, EMRx).
- **Performance** – The method outperforms its baselines (nTsM) as well as the existing methods (EJ66).
- **Application potential** – The method has potential applications in various fields (EJ66).

During the rebuttal phase, two reviewers (**EJ66** and **EMRx**) maintained their scores at 5 (recommend acceptance). Reviewer **nTsM** decided to raise the score from 3 to a positive rating (>=4), acknowledging that most of their concerns had been addressed. Reviewer **EMRx** expressed an overall positive impression of our method in the initial review and explicitly stated: *"I am willing to raise my score if my concerns are well addressed."* In our rebuttal, we devoted substantial effort to conducting the additional experiments as requested and suggested by the reviewers. The new experimental results further demonstrate the advantages of our method, and we believe we have satisfactorily addressed the raised concerns.

We sincerely thank the ACs for facilitating the review process and all reviewers for their insightful and constructive feedback. We will incorporate their suggestions to further improve our manuscript.

Best regards,

The Authors

---

### Decision · Program_Chairs · 2025-09-17

**Decision:**

Reject

**Comment:**

The proposed method offers limited technical novelty as it primarily constitutes a straightforward extension of Noise2Noise (introduced prior to 2020). While the authors have addressed most of the concerns raised during rebuttal, several critical issues remain unresolved. Notably, the method depends heavily on a pretrained model, which introduces an unfair comparison with baseline approaches that do not leverage such prior knowledge. Additionally, the manuscript lacks a thorough analysis of how the choice of the pretrained model influences performance, leaving a key aspect of the methodology inadequately justified. Given these concerns, the manuscript currently does not meet the standard for acceptance.